# ADAPTIVE FOURIER NEURAL OPERATORS: EFFICIENT TOKEN MIXERS FOR TRANSFORMERS

**John Guibas**[3*], **Morteza Mardani**[1*], **Zongyi Li**[1,2], **Andrew Tao**[1],
**Anima Aanandkumar**[1,2], **Bryan Catanzaro**[1]
NVIDIA[1], California Institute of Technology[2], Stanford University[3]
jtguibas@stanford.edu,
{mmardani,zongyil,atao,bcatanzaro,aanandkumar}@nvidia.com

## ABSTRACT

Vision transformers have delivered tremendous success in representation learning. This is primarily due to effective token mixing through self-attention. However, this scales quadratically with the number of pixels, which becomes infeasible for high-resolution inputs. To cope with this challenge, we propose Adaptive Fourier Neural Operator (AFNO) as an efficient token mixer that learns to mix in the Fourier domain. AFNO is based on a principled foundation of operator learning which allows us to frame token mixing as a continuous global convolution without any dependence on the input resolution. This principle was previously used to design FNO, which solves global convolution efficiently in the Fourier domain and has shown promise in learning challenging PDEs. To handle challenges in visual representation learning such as discontinuities in images and high resolution inputs, we propose principled architectural modifications to FNO which results in memory and computational efficiency. This includes imposing a block-diagonal structure on the channel mixing weights, adaptively sharing weights across tokens, and sparsifying the frequency modes via soft-thresholding and shrinkage. The resulting model is highly parallel with a quasi-linear complexity and has linear memory in the sequence size. AFNO outperforms self-attention mechanisms for few-shot segmentation in terms of both efficiency and accuracy. For Cityscapes segmentation with the Segformer-B3 backbone, AFNO can handle a sequence size of 65k and outperforms other self-attention mechanisms. Code is available[1].

## 1 INTRODUCTION

Vision transformers have recently shown promise in producing rich contextual representations for recognition and generation tasks. However, a major challenge is posed by long sequences from high resolution images and videos. Here, long-range and multiway dependencies are crucial to understand the compositionality and relationships among the objects in a scene. A key component for the effectiveness of transformers is attributed to proper mixing of tokens. Finding a good mixer is however challenging as it needs to scale with the sequence size, and systematically generalize to downstream tasks.

Recently, there has been extensive research to find good token mixers; see e.g., Tay et al. (2020b) and references therein. The original self-attention imposes *graph structures*, and uses the similarity among the tokens to capture

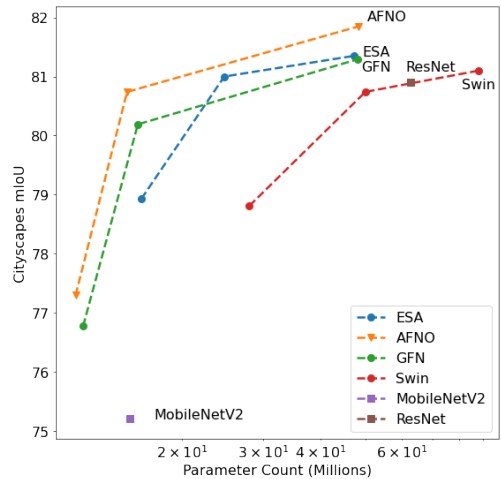

Figure 1: Parameter count and mIoU for Segformer, Swin, and other models at different scales. AFNO consistently outperforms other mixers (see Section 5.7).

---

*Joint first authors, contributed equally. The first author has done this work during internship at NVIDIA, and the second author was leading the project. [1] Code: github.com/jtguibas/AdaptiveFourierNeuralOperator.

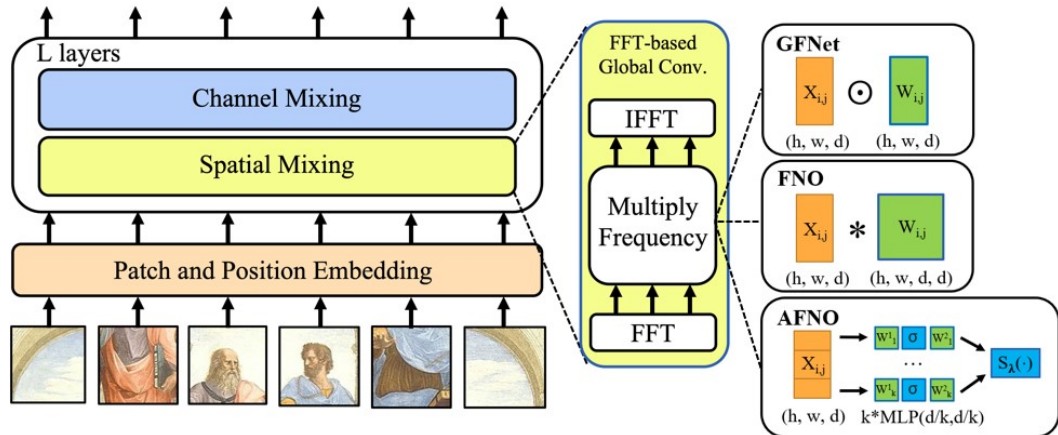

Figure 2: The multi-layer transformer network with FNO, GFN, and AFNO mixers. GFNet performs element-wise matrix multiplication with separate weights across channels ($k$). FNO performs full matrix multiplication that mixes all the channels. AFNO performs block-wise channel mixing using MLP along with soft-thresholding. The symbols $h$, $w$, $d$, and $k$ refer to the height, width, channel size, and block count, respectively.

| Models | Complexity (FLOPs) | Parameter Count | Interpretation |
|---|---|---|---|
| Self-Attention | $N^2 d + 3 N d^2$ | $3d^2$ | Graph Global Conv. |
| GFN | $Nd + Nd \log N$ | $Nd$ | Depthwise Global Conv. |
| FNO | $Nd^2 + Nd \log N$ | $Nd^2$ | Global Conv. |
| AFNO (ours) | $Nd^2/k + Nd \log N$ | $(1 + 4/k)d^2 + 4d$ | Adaptive Global Conv. |

Table 1: Complexity, parameter count, and interpretation for FNO, AFNO, GFN, and Self-Attention. $N := hw$, $d$, and $k$ refer to the sequence size, channel size, and block count, respectively.

the long-range dependencies Vaswani et al. (2017); Dosovitskiy et al. (2020) . It is parameter efficient and adaptive, but suffers from a quadratic complexity in the sequence size. To achieve efficient mixing with linear complexity, several approximations have been introduced for self-attention; see Section 2. These approximations typically compromise accuracy for the sake of efficiency. For instance, long-short (LS) transformer aggregates a long-range attention with dynamic projection to model distant correlations and a short-term attention to capture local correlations Zhu et al. (2021). Long range dependencies are modeled in low dimensions, which can limit expressiveness.

More recently, alternatives have been introduced for self-attention that relax the graph assumption for efficient mixing. Instead, they leverage the *geometric structures* using Fourier transform Rao et al. (2021); Lee-Thorp et al. (2021). For instance, the Global Filter Networks (GFN) proposes depthwise global convolution for token mixing that enjoys an efficient implementation in the Fourier domain Rao et al. (2021). GFN mainly involves three steps: ($i$) spatial token mixing via fast Fourier transform (FFT); ($ii$) frequency gating; and ($iii$) inverse FFT for token demixing. GFN however lacks adaptivity and expressiveness at high resolutions since the parameter count grows with the sequence size, and no channel mixing is involved in ($ii$).

**Our Approach**. To address these shortcomings, we frame token mixing as operator learning that learns mappings between continuous functions in infinite dimensional spaces. We treat tokens as continuous elements in the function space, and model token mixing as continuous global convolution, which captures global relationships in the geometric space. One way to solve global convolution efficiently is through FFT. More generally, we compose such global convolution operations with nonlinearity such as ReLU to learn any general non-linear operator. This forms the basis for designing Fourier Neural operators (FNOs) which has shown promise in solving PDEs Li et al. (2020a). We thus adopt FNO as a starting point for designing efficient token mixing.

**Designing AFNO**. Adapting FNO from PDEs to vision needs several design modifications. Images have high-resolution content with discontinuities due to edges and other structures. The channel mixing in standard FNO incurs a quadratic complexity in the channel size. To control this com-

plexity, we impose a block-diagonal structure on the channel mixing weights. Also, to enhance generalization, inspired by sparse regression, we sparsify the frequencies via soft-thresholding Tibshirani (1996). Also, for parameter efficiency, our MLP layer shares weights across tokens (see Table 1). We term the resulting model as adaptive FNO (AFNO).

We perform extensive experiments with pretraining vision transformers for upstream classification and inpainting that are then finetuned for downstream segmentation. Compared with the state-of-the-art, our AFNO using the ViT-B backbone outperforms existing GFN, LS, and self-attention for few-shot segmentation in terms of both efficiency and accuracy, e.g, compared with self-attention, AFNO achieves slightly better accuracy while being 30% more efficient. For Cityscapes segmentation with the Segformer-B3 backbone, AFNO achieves state-of-the-art and beats previous methods, e.g. AFNO achieves more than 2% better mIoU compared with efficient self-attention Xie et al. (2021), and is also competitive with GFN and LS.

**Key Contributions**. Our main contributions are summarized as follows:

- We establish a link between operator learning and high-resolution token mixing and adapt FNO from PDEs as an efficient mixer with a quasi-linear complexity in the sequence length.
- We design AFNO in a principled way to improve its expressiveness and generalization by imposing block-diagonal structure, adaptive weight-sharing, and sparsity.
- We conduct experiments for pretraining and finetuning. AFNO outperforms existing mixers for few-shot segmentation. For Cityscapes segmentation with the Segformer-B3 backbone, AFNO (sequence: 65k) achieves state-of-the-art, e.g., with 2% gain over the efficient self-attention.

## 2 RELATED WORKS

Our work is at the intersection of operator learning and efficient transformers. Since the inception of transformers, there have been several works to improve the efficiency of self-attention. We divide them into three lines of work based on the structural constraints.

**Graph-Based Mixers** primarily focus on finding efficient surrogates to approximate self-attention. Those include: $(i)$ sparse attentions that promote predefined sparse patterns; see e.g., sparse transformer Child et al. (2019), image transformer Parmar et al. (2018), axial transformer Ho et al. (2019), and longformer Beltagy et al. (2020); $(ii)$ low-rank attention that use linear sketching such as linformers Wang et al. (2020), long-short transformers Lian et al. (2021), Nyströmformer Xiong et al. (2021); $(iii)$ kernel methods that approximate attention with ensemble of kernels such as performer Choromanski et al. (2020), linear transformer Katharopoulos et al. (2020), and random feature attention Peng et al. (2021); and $(iv)$ clustering-based methods such as reformer Kitaev et al. (2020), routing transformer Roy et al. (2021), and Sinkhorn transformer Tay et al. (2020a). These surrogates however compromise accuracy for efficiency.

**MLP-Based Mixers** relax the graph similarity constraints of the self-attention and spatially mix tokens using MLP projections. The original MLP-mixer Tolstikhin et al. (2021) achieves similar accuracy as self-attention. It is further accelerated by ResMLP Touvron et al. (2021) that replaces the layer norm with the affine transforms. gMLP Liu et al. (2021a) also uses an additional gating to weight tokens before mixing. This class of methods however lack scalability due to quadratic complexity of MLP projection, and their parameter inefficiency for high resolution images.

**Fourier-Based Mixers** apply the Fourier transform to spatially mix tokens. FNet Lee-Thorp et al. (2021) resembles the MLP-mixer with token mixer simply being pre-fixed DFT. No filtering is done to adapt the data distribution. Global filter networks (GFNs) Rao et al. (2021) however learn Fourier filters to perform depthwise global convolution, where no channel mixing is involved. Also, GFN filters lack adaptivity that could negatively impact generalization. In contrast, our proposed AFNO performs global convolution with dynamic filtering and channel mixing that leads to better expressivity and generalization.

**Operator Learning** deals with mapping from functions to functions and commonly used for PDEs. Operator learning can be deployed in computer vision as images are RGB-valued functions on a 2D plane. This continuous generalization allows us to permeate benefits from operators. Recent advances in operator learning include DeepONet Lu et al. (2019) that learns the coefficients and basis of the operators, and neural operators Kovachki et al. (2021) that are parameterized by integral

operators. In this work, we adopt Fourier neural operators Li et al. (2020a) that implement global convolution via FFT which has been very successful for solving nonlinear and chaotic PDEs.

# 3 PRELIMINARIES AND PROBLEM STATEMENT

Consider a 2D image that is divided into a $h \times w$ grid of small and non-overlapping patches. Each patch is represented as a $d$-dimensional token, and the image can be represented as a token tensor $X \in \mathbb{R}^{h \times w \times d}$. Treating image as a token sequence, transformers then aim to learn a contextual embedding that transfers well to downstream tasks. To end up with a rich representation, the tokens need to be effectively mixed over the layers.

Self-attention is an effective mixing that learns the graph similarity among tokens. It however scales quadratically with the sequence size, which impedes training high resolution images. Our goal is then to find an alternative mixing strategy that achieves favorable scaling trade-offs in terms of computational complexity, memory, and downstream transfer accuracy.

## 3.1 KERNEL INTEGRATION

The self-attention mechanism can be written as a kernel integration (Tsai et al., 2019; Cao, 2021; Kovachki et al., 2021). For the input tensor $X$ we denote the $(n, m)$-th token as $x_{n,m} \in \mathbb{R}^d$. For notation convenience, we index the token sequence as $X[s] := X[n_s, m_s]$ for some $s, t \in [hw]$. Define also $N := hw$ as the sequence length. The self-attention mixing is then defined as follows:

**Definition 1 (Self Attention).** $\text{Att} : \mathbb{R}^{N \times d} \to \mathbb{R}^{N \times d}$

$$\text{Att}(X) := \text{softmax}\left(\frac{XW_q(XW_k)^\top}{\sqrt{d}}\right) XW_v \tag{1}$$

where $W_q, W_k, W_v \in \mathbb{R}^{d \times d}$ are the query, key, and value matrices, respectively. Define $K := \text{softmax}(\langle XW_q, XW_k \rangle / \sqrt{d})$ as the $N \times N$ score array with $\langle \cdot, \cdot \rangle$ being inner product in $\mathbb{R}^d$. We then treat self-attention as an asymmetric matrix-valued kernel $\kappa : [N] \times [N] \to \mathbb{R}^{d \times d}$ parameterized as $\kappa[s, t] = K[s, t] \cdot W_v$ (where $K[s, t]$ is scalar valued and "·" is scalar-matrix multiplication). Then the self-attention can be viewed as a kernel summation.

$$\text{Att}(X)[s] := \sum_{t=1}^{N} X[t]\kappa[s, t] \qquad \forall s \in [N]. \tag{2}$$

Where $X[t]\kappa[s, t] = X[t]K[s, t]W_v = K[s, t]X[t]W_v$. This kernel summation can be extended to continuous kernel integrals. The input tensor $X$ is no longer a finite-dimensional vector in the Euclidean space $X \in \mathbb{R}^{N \times d}$, but rather a spatial function in the function space $X \in (D, \mathbb{R}^d)$ defined on domain $D \subset \mathbb{R}^2$ which is the physical space of the images. In this continuum formulation, the neural network becomes an operator that acts on the input functions. This brings us efficient characterization originating from operator learning.

**Definition 2 (Kernel Integral).** We define the kernel integral operator $\mathcal{K} : (D, \mathbb{R}^d) \to (D, \mathbb{R}^d)$ as

$$\mathcal{K}(X)(s) = \int_D \kappa(s, t)X(t)\, dt \qquad \forall s \in D. \tag{3}$$

with a continuous kernel function $\kappa : D \times D \to \mathbb{R}^{d \times d}$ Li et al. (2020b). For the special case of the Green's kernel $\kappa(s, t) = \kappa(s - t)$, the integral leads to global convolution defined below.

**Definition 3 (Global Convolution).** Assuming $\kappa(s, t) = \kappa(s - t)$, the kernel operator admits

$$\mathcal{K}(X)(s) = \int_D \kappa(s - t)X(t)\, dt \qquad \forall s \in D. \tag{4}$$

The convolution is a smaller complexity class of operation compared to integration. The Green's kernel has beneficial regularization effect but it is also expressive enough to capture global interactions. Furthermore, the global convolution can be efficiently implemented by the FFT.

## 3.2 Fourier Neural Operator as Token Mixer

The class of shift-equivariant kernels has a desirable property that they can be decomposed as a linear combination of eigen functions. Eigen transforms have a magical property where according to the convolution theorem Soliman & Srinath (1990), global convolution in the spatial domain amounts to multiplication in the eigen transform domain. A popular example of such eigen functions is the Fourier transform. Accordingly, one can define the Fourier neural operator (FNO) Li et al. (2020a).

**Definition 4 (Fourier Neural Operator).** For the continuous input $X \in D$ and kernel $\kappa$, the kernel integral at token $s$ is found as

$$\mathcal{K}(X)(s) = \mathcal{F}^{-1}\big(\mathcal{F}(\kappa) \cdot \mathcal{F}(X)\big)(s) \qquad \forall s \in D,$$

where $\cdot$ denotes matrix multiplication, and $\mathcal{F}, \mathcal{F}^{-1}$ denote the continous Fourier transform and its inverse, respectively.

**Discrete FNO.** Inspired by FNO, for images with finite dimension on a discrete grid, our idea is to mix tokens using the discrete Fourier transform (DFT). For the input token tensor $X \in \mathbb{R}^{h \times w \times d}$, define the complex-valued weight tensor $W := \mathrm{DFT}(\kappa) \in \mathbb{C}^{h \times w \times d \times d}$ to parameterize the kernel. FNO mixing then entails the following operations per token $(m, n) \in [h] \times [w]$

step (1). token mixing $\qquad\qquad z_{m,n} = [\mathrm{DFT}(X)]_{m,n}$

step (2). channel mixing $\qquad\qquad \tilde{z}_{m,n} = W_{m,n} z_{m,n}$

step (3). token demixing $\qquad\qquad y_{m,n} = [\mathrm{IDFT}(\tilde{Z})]_{m,n}$

**Local Features.** DFT assumes a global convolution applied on periodic images, which is not typically true for real-world images. To compensate local features and non-periodic boundaries, we can add a residual term $x_{m,n}$ (can also be parameterized as a simple local convolution) to the token demixing step 3 in FNO; see also Wen et al. (2021).

**Resolution Invariance.** The FNO model is invariant to the discretization $h, w$. It parameterizes the tokens function via Fourier bases which are invariant to the underlying resolution. Thus, after training on one resolution it can be directly evaluated at another resolution (zero-shot super-resolution). Further, the FNO model encodes the higher-frequency information in the channel dimension. Thus, even after truncating the higher frequency modes $z_{m,n}$, FNO can still output the full spectrum.

It is also important to recognize step (2) of FNO, where $d \times d$ weight matrix $W_{m,n}$ mixes the channels. This implies mixed-channel global convolution. Note that the concurrent GFN work Rao et al. (2021) is a special case of FNO, when $W_{m,n}$ is diagonal and the channels are separable.

The FNO incurs $O(N \log(N) d^2)$ complexity, and thus quasi-linear in the sequence size. The parameter count is however $O(N d^2)$ as each token has its own channel mixing weights, which poorly scales with the image resolution. In addition, the weights $W_{m,n}$ are static, which can negatively impact the generalization. The next section enhances FNO to cope with these shortcomings.x

## 4 Adaptive Fourier Neural Operators for Transformers

This section fixes the shortcomings of FNO for images to improve scalability and robustness.

**Block-Diagonal Structure on $W$.** FNO involves $d \times d$ weight matrices for each token. That results in $O(N d^2)$ parameter count that could be prohibitive. To reduce the paramater count we impose a block diagonal structure on $W$, where it is divided into $k$ weight blocks of size $d/k \times d/k$. The kernel then operates independently on each block as follows

$$\tilde{z}_{m,n}^{(\ell)} = W_{m,n}^{(\ell)} z_{m,n}^{(\ell)}, \quad \ell = 1, \ldots, k \tag{5}$$

The block diagonal weights are both interpretable and computationally parallelizable. In essence, each block can be interpreted a head as in multi-head self-attention, which projects into a subspace of the data. The number of blocks should be chosen properly so each subspace has a sufficiently large dimension. In the special case, that the block size is one, FNO coincides with the GFN kernels. Moreover, the multiplications in (5) are performed independently, which is quite parallelizable.

```
def AFNO(x)                                     x = Tensor[b, h, w, d]
  bias = x                                      W_1, W_2 = ComplexTensor[k, d/k, d/k]
  x = RFFT2(x)                                  b_1, b_2 = ComplexTensor[k, d/k]
  x = x.reshape(b, h, w//2+1, k, d/k)
  x = BlockMLP(x)
  x = x.reshape(b, h, w//2+1, d)                def BlockMLP(x):
  x = SoftShrink(x)                               x = MatMul(x, W_1) + b_1
  x = IRFFT2(x)                                    x = ReLU(x)
  return x + bias                                 return MatMul(x, W_2) + b_2
```

Figure 3: Pseudocode for AFNO with adaptive weight sharing and adaptive masking.

**Weight Sharing**. Another caveat with FNO is that the weights are static and once learned they will not be adaptively changed for the new samples. Inspired by self-attention we want the tokens to be adaptive. In addition, static weights are independent across tokens, but we want the tokens interact and decide about passing certain low and high frequency modes. To this end, we adopt a two-layer perceptron that is supposed to approximate any function for a sufficiently large hidden layer. For $(n, m)$-th token, it admits

$$\tilde{z}_{m,n} = \text{MLP}(z_{m,n}) = W_2\sigma(W_1 z_{m,n}) + b \tag{6}$$

Note, that the weights $W_1, W_2, b$ are shared for all tokens, and thus the parameter count can be significantly reduced.

**Soft-Thresholding and Shrinkage**. Images are inherently sparse in the Fourier domain, and most of the energy is concentrated around low frequency modes. Thus, one can adaptively mask the tokens according to their importance towards the end task. This can use the expressivity towards representing the important tokens. To sparsify the tokens, instead of linear combination as in (5), we use the nonlinear LASSO Tibshirani (1996) channel mixing as follow

$$\min \|\tilde{z}_{m,n} - W_{m,n}z_{m,n}\|^2 + \lambda\|\tilde{z}_{m,n}\|_1 \tag{7}$$

This can be solved via soft-thresholding and shrinkage operation

$$\tilde{z}_{m,n} = S_\lambda(W_{m,n}z_{m,n}) \tag{8}$$

that is defined as $S_\lambda(x) = \text{sign}(x)\max\{|x| - \lambda, 0\}$, where $\lambda$ is a tuning parameter that controls the sparsity. It is also worth noting that the promoted sparsity can also regularize the network and improve the robustness.

With the aforementioned modifications, the overall AFNO mixer module is shown in Fig 1 along with the pseudo code in Fig 2. Also, for the sake of comparison, the AFNO is compared against FNO, GFN, and self-attention in Table 1 in terms of interpretation, memory, and complexity.

## 5 EXPERIMENTS

We conduct extensive experiments to demonstrate the merits of our proposed AFNO transformer. Namely, 1) we evaluate the efficiencyy-accuracy trade-off between AFNO and alternative mixing mechanisms on inpainting and classification pretraining tasks; and then 2) measure performance on few-shot semantic segmentation with inpainting pretraining; and 3) evaluate the performance of AFNO in high resolution settings with semantic segmentation. Our experiments cover a wide-range of datasets, including ImageNet-1k, CelebA-Faces, LSUN-Cats, ADE-Cars, and Cityscapes as in Deng et al. (2009); Liu et al. (2015); Yu et al. (2015); Krause et al. (2013); Cordts et al. (2016).

### 5.1 IMAGENET-1K INPAINTING

We conduct image inpainting experiments which compare AFNO to other competitive mixing mechanisms. The image inpainting task is defined as follows: given an input image $X$ of size $[h, w, d]$, where $h, w, d$ denote height, width, and channels respectively, we randomly mask pixel intensities to zero based on a uniformly random walk. The loss function used to train the model is mean squared error between the original image and the reconstruction. We measure performance via the Peak Signal-to-Noise Ratio (PSNR) and structural similarity index measure (SSIM) between the ground truth and the reconstruction. More details about the experiments are provided in the appendix.

| Backbone | Mixer | Params | GFLOPs | Latency(sec) | SSIM | PSNR(dB) |
|----------|-------|--------|--------|--------------|------|----------|
| ViT-B/4 | Self-Attention | 87M | 357.2 | 1.2 | **0.931** | **27.06** |
| ViT-B/4 | LS | 87M | 274.2 | 1.4 | 0.920 | 26.18 |
| ViT-B/4 | GFN | 87M | 177.8 | 0.7 | 0.928 | 26.76 |
| ViT-B/4 | AFNO (ours) | 87M | 257.2 | 0.8 | **0.931** | 27.05 |

Table 2: Inpainting PSNR and SSIM for ImageNet-1k validation data. AFNO matches the performance of Self-Attention despite using significantly less FLOPs.

**Inpainting Results.** PSNR and SSIM are reported in Table 2 for AFNO versus alternative mixers. It appears that AFNO is competitive with self-attention. However, AFNO uses significantly less GFLOPs than Self-Attention. Compared to both LS and GFN, AFNO acheives significantly better PSNR and SSIM. More importantly, AFNO achieves favorable downstream transfer, which is elaborated in the next section for few-shot segmentation.

## 5.2 FEW SHOT SEGMENTATION

After pretraining on image inpainting, we evaluate the few-shot semantic segmentation performance of the models. We construct three few-shot segmentation datasets by selecting training and validation images from CelebA-Faces, ADE-Cars, and LSUN-Cats as in Zhang et al. (2021b). The model is trained using cross-entropy loss. We measure mIoU over the validation set. More details about the experiments are deferred to the Appendix.

| Backbone | Mixer | Params | GFLOPs | LSUN-Cats | ADE-Cars | CelebA-Faces |
|----------|-------|--------|--------|-----------|----------|--------------|
| ViT-B/4 | Self-Attention | 87M | 357.2 | 35.57 | 49.26 | **56.91** |
| ViT-B/4 | LS | 87M | 274.2 | 20.29 | 29.66 | 41.36 |
| ViT-B/4 | GFN | 87M | 177.8 | 34.52 | 47.84 | 55.21 |
| ViT-B/4 | AFNO (ours) | 87M | 257.2 | **35.73** | **49.60** | 55.75 |

Table 3: Few-shot segmentation mIoU for AFNO versus alternative mixers. AFNO surpasses Self-Attention for 2/3 datasets while using less flops.

**Few-Shot Segmentation Results.** Results are reported in Table 3. It is evident that AFNO performs on par with self-attention. Furthermore, for out-of-domain datasets such as ADE-Cars or LSUN-Cats it slightly outperforms self-attention, which is partly attributed to the sparsity regularization endowed in AFNO.

## 5.3 CITYSCAPES SEGMENTATION

To test the scalability of AFNO for high resolution images with respect to alternative mixers, we evaluate high-resolution ($1024 \times 1024$) semantic segmentation for the Cityscapes dataset. We use the SegFormer-B3 backbone which is a hiearchical vision transformer Xie et al. (2021). We train the model using the cross-entropy loss and measure performance via reporting mIoU over the validation set. More details about the experiments and the model are available in the Appendix.

| Backbone | Mixer | Params | Total GFLOPs | Mixer GFLOPs | mIoU |
|----------|-------|--------|--------------|--------------|------|
| Segformer-B3/4 | SA | 45M | N/A | 825.7 | N/A |
| Segformer-B3/4 | Efficient SA | 45M | 380.7 | 129.9 | 79.7 |
| Segformer-B3/4 | LS | 45M | 409.1 | 85.0 | 80.5 |
| Segformer-B3/4 | GFN | 45M | 363.4 | 2.6 | 80.4 |
| Segformer-B3/4 | AFNO-100% (ours) | 45M | 440.0 | 23.7 | **80.9** |
| Segformer-B3/4 | AFNO-25% (ours) | 45M | 429.0 | 12.4 | 80.4 |

Table 4: mIoU and FLOPs for Cityscapes segmentation at $1024 \times 1024$ resolution. Note, both the mixer and total FLOPs are included. For GFN and AFNO, the MLP layers are the bottleneck for the complexity. Also, AFNO-25% only keeps 25% of the low frequency modes, while AFNO-100% keeps all the modes. Results for self-attention cannot be obtained due to the long sequence length in the first few layers.

**Cityscapes Segmentation Results.** We report the final numbers for Cityscapes semantic segmentation in Table 4. AFNO-100% outperforms all other methods in terms of mIoU. Furthermore, we find that the AFNO-25% model which truncates 75% of high frequency modes during finetuning only loses 0.05 mIoU and is competitive with the other mixers. It is important to note that the majority of computations is spent for the MLP layers after the attention module.

## 5.4 IMAGENET-1K CLASSIFICATION

We run image classification experiments with the AFNO mixer module using the ViT backbone on ImageNet-1K dataset containing 1.28M training images and 50K validation images from $1,000$ classes at $224 \times 224$ resolution. We measure performance via reporting top-1 and top-5 validation accuracy along with theoretical FLOPs of the model. More details about the experiments are provided in the appendix.

| Backbone | Mixer | Params | GFLOPs | Top-1 Accuracy | Top-5 Accuracy |
|----------|-------|--------|--------|----------------|----------------|
| ViT-S/4 | LS | 16M | 15.8 | 80.87 | 95.31 |
| ViT-S/4 | GFN | 16M | 6.1 | 78.77 | 94.4 |
| ViT-S/4 | AFNO (ours) | 16M | 15.3 | **80.89** | **95.39** |

Table 5: ImageNet-1K classification efficiencyy-accuracy trade-off when the input resolution is $224 \times 224$.

**Classification Results.** The classification accuracy for different token mixers are listed in Table 5. It can be observed that AFNO outperforms GFN by more than $2\%$ top-1 accuracy thanks to the adaptive weight sharing which allows for a larger channel size. Furthermore, our experiments demonstrate that AFNO is competitive with LS for classification.

## 5.5 ABLATION STUDIES

We also conduct experiments to investigate how different components of AFNO contribute to performance.

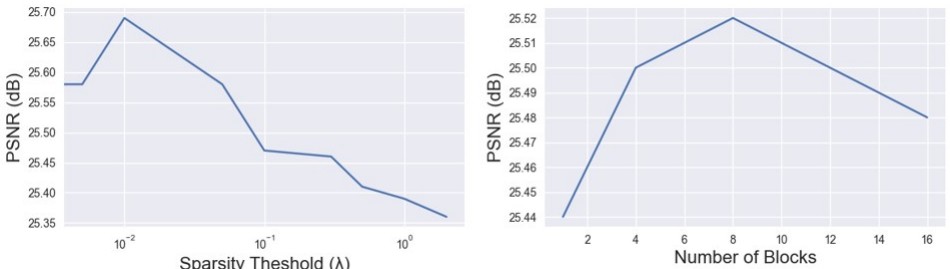

Figure 4: Ablations for the sparsity thresholds and block count measured by inpainting validation PSNR. The results suggest that soft thresholding and blocks are effective

**Sparsity Threshold.** We vary the sparsity threshold $\lambda$ from 0 to 10. For each $\lambda$ we pretrain the network first, and then finetune for few-shot segmentation on the CelebA-Faces dataset. We report both the inpainting PSNR from pretraining and the segmentation mIoU. The results are shown in Figure 3. $\lambda = 0$ corresponds to no sparsity. It is evident that the PSNR/mIoU peaks at $\lambda = 0.01$, indicating that the sparsity is effective. We also compare to hard thresholding (always removing higher frequencies as in FNO) in Table 6. We truncate 65% of the higher frequencies for both inpainting pretraining and few-shot segmentation finetuning.

**Number of Blocks.** We vary the number of blocks used when we impose structure on our weights $W$. To make the comparison fair, we simultaneously adjust the hidden size so the overall parameter count of the model are equal. We vary the number of blocks from 1 to 64 and measure the resulting inpainting PSNR on ImageNet-1K. It is seen that 8 blocks achieves the best PSNR. This shows that blocking is effective.

**Impact of Adaptive Weights.** We evaluate how removing adaptive weights and instead using static weights affects the performance of AFNO for ImageNet-1K inpainting and few-shot segmentation.

The results are presented in Table 6. The results suggest that adaptive weights are crucial to AFNO's performance.

| Backbone | Mixer | Parameter Count | PSNR | CelebA-Faces mIoU |
|---|---|---|---|---|
| ViT-XS/4 | FNO | 16M | 24.8 | 39.27 |
| ViT-XS/4 | AFNO [Non-Adaptive Weights] | 16M | 25.1 | 44.04 |
| ViT-XS/4 | AFNO [Hard Thresholding 35%] | 16M | 23.58 | 34.17 |
| ViT-XS/4 | AFNO | 16M | **25.69** | **49.49** |

Table 6: Ablations for AFNO versus FNO, AFNO without adaptive weights, and hard thresholding. Results are on inpainting pretraining with 10% of ImageNet along with few-show segmentation mIoU on CelebA-Faces. Hard thresholding only keeps 35% of low frequency modes. AFNO demonstrates superior performance for the same parameter count in both tasks.

**Comparison to FNO.** To show that AFNO's modifications fix the shortcomings of FNO for images, we directly compare AFNO and FNO on ImageNet-1K inpainting pretraining and few-shot segmentation on CelebA-Faces. The results are also presented in Table 6. The results suggest that AFNO's modifications are crucial to performance in both tasks.

## 5.6 COMPARISON WITH DIFFERENT TRUNKS AT DIFFERENT SCALES

In order to provide more extensive comparison with the state-of-the-art efficient transformers we have included experiments for different trunks at different scales. Since the primary motivation of this work is to deal with high resolution vision, we focus on the task of Cityscapes semantic segmentation a the benchmark that is a challenging task due to the high 1024×2048 resolution of images. For the trunks we adopt: i) the Segformer Xie et al. (2021) backbones B0, B1, B2, B3, under three different mixers namely AFNO, GFN and efficient self-attention (ESA); ii) Swin backbones Liu et al. (2021b) (T, S, B), and iii) ResNet He et al. (2016) and MobileNetV2 Sandler et al. (2018). Results for LS and self-attention are not reported due to instability issues with half-precision training and quadratic memory usage with sequence length respectively. Note that efficient self-attention is self-attention but with a sequence reduction technique introduced in Wang et al. (2021). It is meant to be a cheap approximation to self-attention. Numbers for ResNet and MobileNetV2 are directly adopted from Xie et al. (2021). For training Segformer we use the same recipe as discussed in Section A.3 which consists of pretraining on ImageNet-1K classification for 300 epochs. For training Swin, we use pretrained classification checkpoints available from the original authors and then combine Swin with the Segformer head Xie et al. (2021). We use the same training recipe as the Segformer models for Swin.

The mIoU scores are listed in Fig. 1 versus the parameter size. It is first observed that AFNO outperforms other mixers when using the same Segformer backbone under the same parameter size. Also, when using AFNO with the hierarchical segformer backbone, it consistently outperforms the Swin backbone for semantic segmentation.

## 6 CONCLUSIONS

We leverage the geometric structure of images in order to build an efficient token mixer in comparison to self-attention. Inspired by global convolution, we borrow Fourier Neural Operators (FNO) from PDEs for mixing tokens and propose principled architectural modifications to adapt FNO for images. Specifically, we impose a block diagonal structure on the weights, adaptive weight sharing, and sparsify the frequency with soft-thresholding and shrinkage. We call the proposed mixer Adaptive Fourier Neural Operator (AFNO) and it incurs quasi-linear complexity in sequence length. Our experiments indicate favorable accuracy-efficiency trade-off for few-shot segmentation, and competitive high-resolution segmentation compared with state-of-the-art. There are still important avenues to explore for the future work such as exploring alterantives for the DFT such as the Wavelet transform to better capture locality as in Gupta et al. (2021).

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

## A  APPENDIX

This section includes visualizations of AFNO as well as the details of the experiments.

### A.1  VISUALIZATION OF AFNO

To gain insight into how AFO works, we produce visualizations of AFNO's weights and representations. In particular, we visualize the spectral clustering of the tokens and sparsity masks from soft thresholding.

**AFNO clustering versus other mixers**. We show the clustering of tokens after each transformer layer. For a 10-layer transformer pretrained with 10% of ImageNet-1k, we apply spectral clustering on the intermediate features when we use k-NN kernel with $k = 10$ and the number of clusters is also set to $4$. It appears that AFNO clusters are as good as self-attention. AFNO clusters seem to be more aligned with the image objects than GFN ones. Also, long-short transformer seems not to preserve the objectness in the last layers. This observation is consistent with the the few-shot segmentation results in section 5.2 that show the superior performance of AFNO over the alternatives.

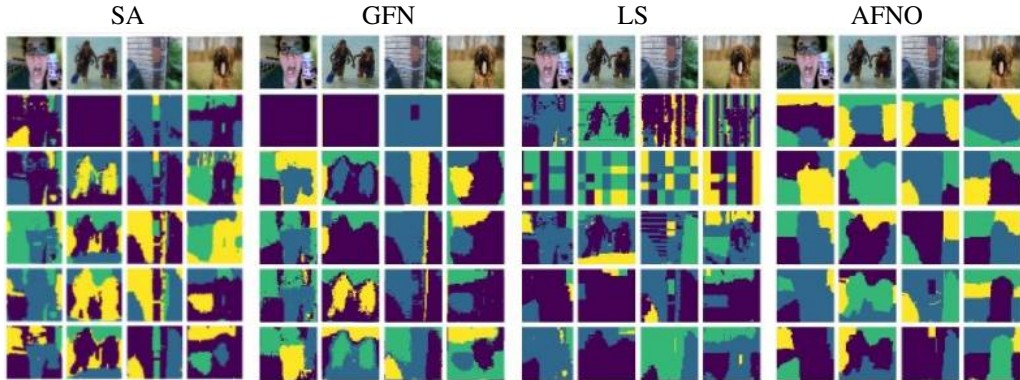

Figure 5: Spectral clustering of tokens for different token mixers. From top to bottom, it shows the input and the layers $2, 4, 6, 8, 10$ for the inpainting pretrained model.

**Sparsity Masks**. We also explore how the sparsity mask affects the magnitude of the values in tokens. We calculate the fraction of values over the channel dimension and blocks that have been masked to zero by the Softshrink function. As one can see in Figure 6, it is clear that the input images are very sparse in the Fourier domain. Furthermore, this sparsity suggests that we can aggressively truncate higher frequencies and maintain performance.

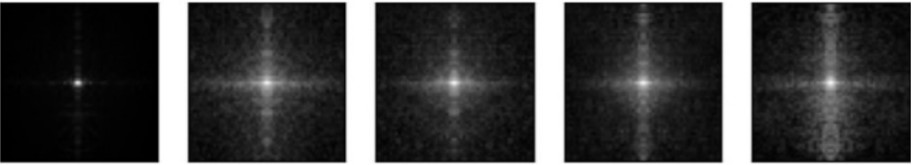

Figure 6: Log magnitude of tokens ($56{\times}56$) after soft-thresholding and shrinkage ($\lambda = 0.1$) and the sparsity mask averaged over channels for inpainting pretrained network. Left to right shows layers 1 to 5, respectively.

### A.2  INPAINTING

We use the ViT-B/4 backbone for the ImageNet-1K inpainting experiments. The ViT-B backbone has 12 layers and is described in more detail in the original paper Dosovitskiy et al. (2020). Importantly, we use a 4x4 patch size to model the long sequence size setting.

- Self-Attention Vaswani et al. (2017) uses 16 attention heads and a hidden size of 768.

- Long-Short transformer Zhu et al. (2021) (LS) uses a window-size of $4$ and dynamic projection rank of $8$ with a hidden size of $768$.

- Global Filter Network Rao et al. (2021) (GFN) uses a hidden size of $768$.

- Adaptive Fourier Neural Operator (AFNO) uses $1$ block, a hidden dimension of $750$, and a sparsity threshold of $0.1$, and a 1D convolution layer as the bias.

The training procedure can be summarized as follows. Given an image $x \in \mathbb{R}^{3 \times 224 \times 224}$ from ImageNet-1K, we randomly mask pixel intensities to zero by initially sampling uniformly from $\{(i,j)\}_{i,j=1}^{h,w}$ and then apply the transition function $T((i,j)) = \text{UniformSample}(\{(i-1,j),(i+1,j),(i,j-1),(i,j+1)\})$ for 3136 steps. We use a linear projection layer at the end to convert tokens into a reconstructed image. Our loss function computes the mean squared error between the ground truth and reconstruction of the masked pixels. We measure performance via the Peak Signal-to-Noise Ratio (PSNR) and structural similarity index measure (SSIM) between the ground truth and the reconstruction.

We train for 100 epochs using the Adam optimizer with a learning rate of $10^{-4}$ for self-attention and $10^{-3}$ for all the other mechanisms using the cosine-decay schedule to a minimum learning rate of $10^{-5}$. We use gradient clipping threshold of 1.0 and weight-decay of 0.01.

## A.3 Cityscapes Segmentation

We use the SegFormer-B3 backbone for the Cityscapes segmentation experiments. The SegFormer-B3 backbone is a four-stage architecture which reduces the sequence size and increase the hidden size as you progress through the network. The model is described in more detail in Xie et al. (2021). More details about how the mixing mechanisms are combined with SegFormer-B3 is described below. All models do not modify the number layers in each stage which is [3, 4, 18, 3].

- Efficient Self-Attention uses a hidden size of [64, 128, 320, 512] and [1, 2, 5, 8] for the four stages

- Global Filter Network uses a hidden size of [128, 256, 440, 512] in order to match the parameter count of the other networks. Because GFN is not resolution invaraint, we use bilinear interpolation in the forward pass to make the filters match the input resolution of the tokens.

- Long-Short uses a hidden size of [128, 256, 360, 512] to match the parameter count of the other networks and [1, 2, 5, 8] attention heads.

- Adaptive Fourier Neural Operator uses [208, 288, 440, 512] to match the parameter count of the other networks. It uses [1, 2, 5, 8] blocks in the four stages.

We pretrain the SegFormer-B3 backbone on ImageNet-1K classification for 300 epochs. Our setup consists of using the Adam optimizer, a learning rate of $10^{-3}$ with cosine decay to $10^{-5}$, weight regularization of 0.05, a batch size of 1024, gradient clipping threshold of 1.0, and learning rate warmup for 6250 iterations. We then finetune these models on Cityscapes for 450 epochs using a learning rate of $1.2 \cdot 10^{-4}$. We train on random 1024x1024 crops and also evaluate at 1024x1024.

## A.4 Few-Shot Segmentation

The models used for few-shot segmentation are described in B.1. We use the inpainting pretrained models and finetune them on few-shot segmentation on CelebA-Faces, ADE-Cars, and LSUN-Cats at 224x224 resolution. The samples for the few-shot dataset are selected as done in DatasetGAN in Zhang et al. (2021a). To train the network, we use the per-pixel cross entropy loss.

We finetune the models on the few-shot datasets for 2000 epochs with a learning rate of $10^{-4}$ for self-attention and $10^{-3}$ for other mixers. We use no gradient clipping or weight decay. We measure validation performance every 100 epochs and report the maximum across the entire training run.

## A.5 CLASSIFICATION

The models for classification are based on the Global Filter Network GFN-XS models but with 4x4 patch size. In particular, we utilize 12 transformer layers and adjust the hidden size and attention-specific hyperparameters to reach a parameter count of 16M. Due to some attention mechanisms not being able to support class tokens, we use global average pooling at the last layer to produce output softmax probabilities for the $1,000$ classes in ImageNet-1k. More details about each of the models is provided below.

- Self-Attention Vaswani et al. (2017) uses 12 attention heads and a hidden size of $324$.
- Long-Short transformer Zhu et al. (2021) (LS) uses a window-size of $4$ and dynamic projection rank of $8$ with a hidden size of $312$.
- Global Filter Network Rao et al. (2021) (GFN) uses a hidden size of $245$. The hidden size is smaller due to the need to make all the models have the same parameter count.
- Adaptive Fourier Neural Operator (AFNO) uses 16 blocks, a hidden dimension of $384$, sparsity threshold of $0.1$, and a 1D convolution layer as the bias.

We trained for 300 epochs with Adam optimizer and cross-entropy loss using the learning rate of $(\mathsf{BatchSize}/512) \times 5 \times 10^{-4}$ for the models. We also use five epochs of linear learning-rate warmup, and after a cosine-decay schedule to the minimum value $10^{-5}$. Along with this, the gradient norm is clipped not to exceed $1.0$ and weight-decay regularization is set to $0.05$.

## A.6 ABLATION

For the ablation studies, we use a backbone we denote ViT-XS which refers to models which only have 5 layers and have attention-specifc hyperparameters adjusted to reach a parameter count of 16M. Details of these models are described below.

- For FNO, we use a hidden size of 64 and five layers to make the parameter count 16M.
- For AFNO with Static Weights, we use a hidden size of 124, four blocks, and a sparsity theshold of 0.01.
- For AFNO-35%, we hard threshold and only keep the bottom 35% frequencies. In practice, this means we keep 32/56 frequncies of the tokens along each spatial dimension. We use a hidden size of 124, four blocks and no sparsity theshold.
- For AFNO, we use a hidden size of 584, four blocks, and a sparsity theshold of 0.01.

These models are inpaint pretrained on a randomly chosen subset of only 10% of ImageNet-1K and trained for 100 epochs. For finetuning, we use the same setup as the few-shot segmentation experiments described in A.4. We only evaluate on the CelebA-Faces dataset.

