# OpenReview forum: "Efficient Token Mixing for Transformers via Adaptive Fourier Neural Operators"
_ICLR.cc/2022/Conference — ICLR 2022 Poster_

### Official Review · Reviewer_PwZa · 2021-10-26

**Correctness:** 4
**Technical Novelty And Significance:** 3
**Empirical Novelty And Significance:** 4
**Recommendation:** 6
**Confidence:** 4

**Main Review:**

The idea of this paper is novel and interesting. My main concern is its experiments: I think that the experiments are insufficient to support the conclusion and demonstrate the effectiveness of the proposed method. This paper should provide extensive experiments and comparisons for image classification, object detection, instance segmentation, and semantic segmentation. The ablation studies should also be conducted on the ImageNet dataset for image classification. These tasks are the most fundamental ones, which can thus reflect the ability of the represent learning of networks.

However, this paper provides results for some uncommon tasks, like image inpainting and few-shot segmentation. Hence, I have reasons to guess that the proposed method is only effective for these carefully chosen tasks. Although this paper provides some results for semantic segmentation and image classification, the experiments for semantic segmentation are only conducted using Segformer-B3 and the experiments for image classification are only conducted using ViT-S. This seems that this paper carefully chooses one model for each task, and the comparison to state-of-the-art transformer models is also missing. Therefore, I think that current experiments are insufficient to demonstrate the effectiveness of the proposed method.


**Summary Of The Paper:**

Vision transformers scale quadratically with the number of pixels. To cope with this challenge, this paper proposes Adaptive Fourier Neural Operator (AFNO) as an efficient token mixer that learns to mix in the Fourier domain. This is achieved by modifying FNO, including imposing a block-diagonal structure on the channel mixing weights, adaptively sharing weights across tokens, and sparsifying the frequency modes via soft-thresholding and shrinkage. The resulting model has a quasi-linear complexity and linear memory in the sequence size.

**Summary Of The Review:**

My main concern is about the insufficient experiments. Please see the above comments!

---

> ### Author Response · Authors · 2021-11-22
> **Response to R3**
>
> We thank the reviewer for the feedback and finding the paper interesting. We have addressed the concerns below.
>
> *The idea of this paper is novel and interesting. My main concern is its experiments: I think that the experiments are insufficient to support the conclusion and demonstrate the effectiveness of the proposed method. This paper should provide extensive experiments and comparisons for image classification, object detection, instance segmentation, and semantic segmentation. The ablation studies should also be conducted on the ImageNet dataset for image classification. These tasks are the most fundamental ones, which can thus reflect the ability of the representation learning of networks.*
> *However, this paper provides results for some uncommon tasks, like image inpainting and few-shot segmentation. Hence, I have reasons to guess that the proposed method is only effective for these carefully chosen tasks. Although this paper provides some results for semantic segmentation and image classification, the experiments for semantic segmentation are only conducted using Segformer-B3 and the experiments for image classification are only conducted using ViT-S. It seems that this paper carefully chooses one model for each task, and the comparison to state-of-the-art transformer models is also missing. Therefore, I think that current experiments are insufficient to demonstrate the effectiveness of the proposed method.*
> - In accordance with the reviewers comment, we have tried our best to run as many experiments as we can during the revision period (21 experiments each using 32 GPUs). Since the primary focus of this work is on efficient mixers for high resolution vision, we adopt the semantic cityscapes segmentation as the benchmark. This is a standard and challenging task.
> We first pretrained for ImageNet-1k classification, and then finetune for 1024x2048 cityscapes segmentation. We compare our AFNO mixer with other mixers at different scales. We also include comparison with efficient transformer trunks such as Swin, and CNN based trucks such as ResNet/MobileNet. The mIoUs in Fig. 6 (new section A. 6) indicate that AFNO consistently outperforms alternative mixers, and can suppress alternative trunks such as Swin for the same parameter size. In the final version of the paper, we will move Fig. 6 and the corresponding description and discussions to the main body of the paper.
> - We agree that it would be ideal to run more experiments on other standard tasks. However, we believe that we have shown the effectiveness of AFNO across a variety of tasks and settings including classification, semantic segmentation, few-shot segmentation, and inpainting. Our primary reason for not including more experiments is due to resource constraints. For example, running an ImageNet-1K classification experiment on a ViT-B/4 model can take up to a week on 32 GPUs.

---

### Official Review · Reviewer_H7fs · 2021-11-01

**Correctness:** 3
**Technical Novelty And Significance:** 3
**Empirical Novelty And Significance:** 2
**Recommendation:** 8
**Confidence:** 3

**Details Of Ethics Concerns:**

Since this study only evaluates the proposed models on widely-used public datasets, I don't think there are any ethics concerns.

**Main Review:**

It's an interesting idea and a good attempt that aims to propose a solution to reduce the computing burden while maintaining the performance of visual transformer models. The motivation of introducing a Fourier Neural operator is reasonable to improve its expressiveness and generalization by imposing block-diagonal structure, adaptive weight-sharing, and sparsity. However, from my view, the main weakness of this study is lacking comparison experiments on general computer vision tasks with state-of-the-art visual transformer models such as image classification, object detection (these tasks have been included in another visual transformer model "Liu, Ze, et al. "Swin transformer: Hierarchical vision transformer using shifted windows." arXiv preprint arXiv:2103.14030 (2021)."). For example, performance of swin transformer on imagenet 1k and 22k are both superior to this paper. According to the advantages shown in this study, it's better to be titled "an improved transformer model for few-shot segmentation" rather than the existing one which seems much more ambitious. When it comes to more details of this paper, the proposed methods perform very similar with or even worse than some existing ones while needing more network parameters (Table 4).


**Summary Of The Paper:**

As is known, vision transformer has been becoming a more and more popular topic in the area of computer vision which is inspired by the success of this deep neural network fashion in other fields. However, the computing burden is a common disadvantage across the majority of transformer models compared with other deep neural networks. From a view of adaptive network operation in transformer, authors proposed to improve its expressiveness and generalization by imposing block-diagonal structure, adaptive weight-sharing, and sparsity.

**Summary Of The Review:**

It's an interesting study that aims to propose a solution to a common challenge in the visual transformer model. However, I can't support its acceptance considering the presented results.

---

> ### Author Response · Authors · 2021-11-22
> **Response to R2**
>
> We thank the reviewer for the feedback and finding the idea interesting. During the revision period, we have run an extensive set of experiments to address the reviewer’s concerns as follows:
>
> *However, from my view, the main weakness of this study is lacking comparison experiments on general computer vision tasks with state-of-the-art visual transformer models such as image classification, object detection (these tasks have been included in another visual transformer model "Liu, Ze, et al. "Swin transformer: Hierarchical vision transformer using shifted windows." arXiv preprint arXiv:2103.14030 (2021)."). For example, performance of swin transformer on imagenet 1k and 22k are both superior to this paper. According to the advantages shown in this study, it's better to be titled "an improved transformer model for few-shot segmentation" rather than the existing one which seems much more ambitious.*
> - In our paper, we proposed a mixer which can be used with any architectural backbone. As a result, choosing the best backbone is not the point of this paper and it is an orthogonal direction to developing efficient token mixers. Having said that, in light of the reviewer’s comments, in the revised paper, **we perform an extensive comparison with other backbones such as Swin and ResNet/MobileNetV2 at different scales for the Cityscapes semantic segmentation (resolution: 1024x2048) task which is a standard benchmark**. We choose this task because the primary motivation of this work is to deal with high resolution. More specifically, we have included Fig. 6 in the new section A. 6, which shows the mIoU versus the parameter count for a variety of Segformer architectures (using different mixers), Swin, and ResNet/MobileNetV2 models. As seen by the data, the AFNO mixer with the hierarchical Segformer backbone consistently outperforms other mixers and trunks for the same parameter count. We will move this figure to the main section of the paper in the final version.
>
> *When it comes to more details of this paper, the proposed methods perform very similar with or even worse than some existing ones while needing more network parameters (Table 4).*
> - We respectfully disagree with the reviewer’s interpretation of Table 4. The AFNO-25% model truncates 75% of the frequencies, so it is expected that it will not achieve the best mIoU. The AFNO-100% model outperforms every other mixer tested by at least 0.4 mIoU for the same parameter count. While we agree that the gain is not significant, we believe it demonstrates how principled modifications to FNO and continuous global convolution can lead to a mixer that is competitive with the state-of-the-art graph-based mixers (e.g., self-attention, long-short).
> - To provide a more extensive comparison of the mixers at different scales, we have added in the revised manuscript Figure 6 which shows the parameter count versus mIoU with the hierarchical Segformer backbone. It can be observed that the AFNO mixer consistently outperforms other mixers for the same parameter count.

---

### Official Review · Reviewer_UFjk · 2021-11-02

**Correctness:** 1
**Technical Novelty And Significance:** 2
**Empirical Novelty And Significance:** 2
**Recommendation:** 6
**Confidence:** 4

**Main Review:**

The paper proposes an alternative mechanism for self-attention in the context of vision transformers. The proposed method reduces the mixing complexity from quadratic to N log N, which could be important for long token sequences, e.g. high-resolution images.

The paper uses a well-established theory of Fourier transform motivating the proposal, which is neat.

It is hard from the paper presentation to compare with other efficient attention methods - the comparison is done using two metrics, accuracy and flops. Therefore it is not possible to judge if the paper achieved the state-of-the-art.

The novelty is somewhat limited. The differences with GFN, i.e. adding MLP and channel mixing, do not seem particularly insightful or significant.

One weakness is that, at least on the chosen applications, the quadratic complexity of attention is not a bottleneck of the networks. At least it is not clear from the paper. For example, table 4 suggests that the mixer takes at most 35% of total compute, with the rest consumed by the point-wise MLP. From this angle, the significance of the work is not convincingly demonstrated.

I find it weird that the paper on vision transformers does not cite the original Dosovitsky et al.

**Summary Of The Paper:**

The paper proposes a new Adaptive Fourier Neural Operator (AFNO) for mixing tokens in visual transformers. The idea is based on Fourier neural operators (FNO) that transform feature flow in the Fourier space. The difference w.r.t. existing FNO is in two modifications. First, the weight matrix is block-diagonal (analog of multi-head) and, second, use of an MLP instead of just linear weighting. The experiments show that the proposed method is competitive and often achieves results as good as with original self-attention (with fewer flops).

**Summary Of The Review:**

Despite limited novelty and significance, I found the work interesting and recommend acceptance.

---

> ### Author Response · Authors · 2021-11-22
> **Response to R1**
>
> We thank the reviewer for the acceptance recommendation and finding the idea neat. We have addressed the concerns as follows:
>
> *It is hard from the paper presentation to compare with other efficient attention methods - the comparison is done using two metrics, accuracy and flops. Therefore it is not possible to judge if the paper achieved the state-of-the-art.*
> - To address this issue, we have run a new set of extensive experiments in the new section A. 6 of the revised paper to compare the AFNO semantic segmentation accuracy for different network parameter counts. We also have included other efficient transformer trunks such as Swin. **As seen in Fig. 6, the Segformer variant with the AFNO mixer consistently outperforms other mixers (using the same Segformer trunk) and other trunks (e.g., Swin and ResNet) for the same parameter count at different network sizes.**
> *The novelty is somewhat limited. The differences with GFN, i.e. adding MLP and channel mixing, do not seem particularly insightful or significant.*
> - We respectfully disagree. First of all, AFNO rigorously links token mixing to continuous global convolution and operator learning for solving PDEs. This connection is insightful which is in contrast with GFN that heuristically relates to **depth-wise global convolution**. The ability to mix the channel dimensions is important for token mixing as seen by mixers such as self-attention.
> - Second, our modifications such as MLP weight-sharing are actually crucial for scaling to arbitrary resolution sizes, which GFN cannot do without manual interpolation of the weights.
> - Third, as shown in Table 2-3, we empirically observed that principled changes to FNO such as block-diagonal structures and sparsity thresholding can significantly improve expressivity and few-shot generalization compared with GFN.
> *One weakness is that, at least on the chosen applications, the quadratic complexity of attention is not a bottleneck of the networks. At least it is not clear from the paper. For example, table 4 suggests that the mixer takes at most 35% of total compute, with the rest consumed by the pointwise MLP. From this angle, the significance of the work is not convincingly demonstrated.*
> - The quadratic complexity of attention is not a bottleneck for the majority of the networks due to the comparisons being done with other efficient self-attention alternatives. Indeed, as seen from the original self-attention FLOPs in Table 4, the quadratic complexity would make it infeasible to run (see mixer GFLOPs in the first row of Table 4). However, using AFNO significantly reduces the mixer FLOPs. Because we wanted to fix the parameter count, we increased the hidden size which increases the total FLOPS, but one can observe that the mixer FLOPs for AFNO is significantly less compared to both Efficient Self-Attention and Long-Short Transformer. Having said that, we agree, as already pointed out in Section 5.3, the main complexity is shifted to MLP layers when using AFNO and other efficient mixers.
> *I find it weird that the paper on vision transformers does not cite the original Dosovitsky et al.*
> - Thank you for pointing this out. We have updated the text and cited the original paper by Dosovitsky et al.

---

> > ### Comment · Reviewer_UFjk · 2021-11-30
> > **thanks for addressing my comments**
> >
> > After reading the reviews, I don't think it is necessary to compare with contemporaneous networks (some of the mentioned ones were published after the submission deadline such as SWIN).
> >
> > I maintain my rating. I am not entirely convinced about novelty, but I think this paper, nevertheless, can be interesting to the community.

---

### Decision · Program_Chairs · 2022-01-20

**Decision:**

Accept (Poster)

**Comment:**

Overall, this paper receives positive reviews. The reviewers find the technical novelty and contributions are significant enough for acceptance at this conference. The authors' rebuttal helps address some issues. The area chair agrees with the reviewers and recommend it be accepted at this conference.